# Fine Mapping and Identification of a Candidate Gene of Downy Mildew Resistance, *RPF2*, in Spinach (*Spinacia oleracea* L.)

**DOI:** 10.3390/ijms232314872

**Published:** 2022-11-28

**Authors:** Shuo Gao, Tiantian Lu, Hongbing She, Zhaosheng Xu, Helong Zhang, Zhiyuan Liu, Wei Qian

**Affiliations:** Institute of Vegetables and Flowers, Chinese Academy of Agricultural Sciences, Beijing 100081, China

**Keywords:** spinach, downy mildew, disease resistance, candidate gene, molecular marker, breeding

## Abstract

Downy mildew is a major threat to the economic value of spinach. The most effective approach to managing spinach downy mildew is breeding cultivars with resistance genes. The resistance allele *RPF2* is effective against races 1–10 and 15 of *Peronospora farinosa f.* sp. *Spinaciae* (*P. effusa*) and is widely used as a resistance gene. However, the gene and the linked marker of *RPF2* remain unclear, which limit its utilization. Herein, we located the *RPF2* gene in a 0.61 Mb region using a BC_1_ population derived from Sp39 (rr) and Sp62 (RR) cultivars via kompetitive allele specific PCR (KASP) markers. Within this region, only one R gene, *Spo12821*, was identified based on annotation information. The amino acid sequence analysis showed that there were large differences in the length of the LRR domain between the parents. Additionally, a molecular marker, RPF2-IN12821, was developed based on the sequence variation in the *Spo12821*, and the evaluation in the BC_1_ population produced a 100% match with resistance/susceptibility. The finding of the study could be valuable for improving our understanding of the genetic basis of resistance against the downy mildew pathogen and breeding resistance lines in the future.

## 1. Introduction

Spinach (*Spinacia oleracea* L.) is an important edible leafy vegetable that is abundant in nutrition and contains lutein, folate, iron, calcium, and vitamins [1]. Over the years, there has been a growing demand for spinach (especially organically produced). China, the United States, and Turkey are the main spinach-producing countries, especially China, where around 91% of spinach is produced [2].

Downy mildew, caused by *Peronospora farinosa f.* sp. *Spinaciae* (*P. effusa*), is perhaps the most widespread and destructive disease of spinach worldwide [3]. The disease can severely damage the quality and yield of spinach production. Downy mildew pathogens have a short latent period of 6–8 days, and the spinach leaves that are infected by downy mildew pathogens have gray sporulation and chlorotic spots on the abaxial surface. Downy mildew is unable to be controlled by specific fungicides, and treatments are costly and limited. Developing stable genetic resistance is the most economical strategy for managing spinach downy mildew [4].

The pathogen was first reported to infect spinach in 1824; at present, there are 19 unique races documented, of which 16 have been discovered during the last 30 years [5,6,7,8,9,10,11,12]. A working group known as the International Working Group on Peronospora (IWGP) has established differential hosts for physiological races of spinach downy mildew by using fixed spinach varieties and officially named new races. Races 18 and 19 were denominated by the IWGP in 2021 [12]. Once new races of *P. effusa* emerged, there may be those that overcome plant resistance. It has been proposed that there may be six genes involved in the resistance to *Peronospora farinosa* (*RPF1–RPF6*) [13] and *RPF1–3* have been genetically characterized [4]. The locus *RPF1* was discovered by a single dominant allele located on chromosome 3. The codominant molecular marker DM1 from an AFLP fragment is nearly 1.7 cM from the *RPF1* locus [14]. The marker 5B14r, designed from putative resistance gene analogs (RAGs), was found to be co-segregated with the marker DM1 [15]. Five genes (*Spo12736*, *Spo12784*, *Spo12903*, *Spo12905*, and *Spo12821*) close to DM1 were predicted as candidate genes for downy mildew resistance through the NBS-LRR structure [16]. The downy mildew resistance loci *RPF1*, *RPF2*, and *RPF3* were mapped to a 1.5 Mb region of chromosome 3, and markers to distinguish the different loci [4]. In addition, the *RPF1* locus was located at 0.34–1.23 Mb on chromosome 3, and *Spo12784*, *Spo12903*, and *Spo12729* were preliminarily identified as candidate genes for *RPF1* based on protein homology comparisons between the resistant and susceptible lines [17]. According to single SNPs and haplotype association analysis, the *RPF1* locus was narrowed to 0.39–1.23 Mb, and *Spo12784*, *Spo12903*, *Spo12905*, and *Spo12821* were reported as the candidate genes [18]. Using genotyping by sequencing (GBS), the downy mildew resistance locus against *Pfs* 16 was localized to a 0.57 Mb region of *RPF3* on chromosome 3, and four genes (*Spo12736*, *Spo12784*, *Spo12908*, and *Spo12821*) were identified as the best candidate genes [19]. Extensive studies about *RPF1* have been conducted [4,15,16,17,18], while *RPF2* has rarely been reported on, which limits its application.

Quantitative trait locus mapping can aid in understanding the complexity of phenotypes [20]. The principle of the analysis is to make an association between the genotypes of markers and the phenotype [21]. In the present study, the susceptible line Sp39 and the resistant inbred line Sp62 were used as the parents to develop the BC_1_ population. The objectives of the present study were to fine-map the *RPF2* locus and identify the candidate genes via developing molecular markers that were designed by aligning the whole-genome resequencing of the parent with the reference genome in the BC_1_ population derived from the crosses of F_1_ and Sp39. The tightly linked markers could be used for MAS in spinach and to pyramid many RPFs to develop durable resistant cultivars.

## 2. Results

### 2.1. Phenotypic and Genetic Analysis

To access the inheritance of resistance to *Pfs* in BC_1_, a backcross was performed with Sp62(RR) and Sp39(rr) as the recurrent male plants. A total of 400 seeds were planted in Beijing after germination, and 226 individuals survived in the BC_1_ population. Then, 226 individuals were evaluated for resistance, according to the standard inoculation protocol [9,10]. In the BC_1_ population, 110 individuals exhibited symptoms of a downy mildew pathogen, while 116 had no obvious symptoms or oospores through microscopic observation. The ratio of resistance: susceptibility fitted to the expected segregation ratio of 1:1 with a chi-square test (Table 1), suggesting that the resistance to *Pfs* in Sp62 was controlled by a single dominant gene.

### 2.2. Fine Mapping of RPF2

A previous study had located the *RPF2* locus in a 1.5 Mb region on chromosome 3. To further narrow down the candidate region of *RPF2* in spinach, parent lines (Sp62 and Sp39) were re-sequenced to obtain more InDel and SNP information in a 1.5 Mb region. A total of eight molecular markers were developed to fine map the *RPF2* locus in the BC_1_ population. Based on the genome information (Table 2), the *RPF2* locus was finally located in a 0.61 Mb interval on chromosome 3, flanked by the markers KMR15-09 (1.11 Mb) and RPF2–IN172 (1.72 Mb) (Figure 1).

### 2.3. Screening of Candidate Genes

A total of 76 genes were identified in the 0.61 Mb candidate region based on the spinach genome (version Sp75) (Appendix A). Among these genes, 64 putative genes in the region were functionally annotated. Fifteen of the sixty-four genes encoded various enzymes, including ten genes that encoded cellular components and fourteen genes that were involved in biological processes. In particular, we found an R gene, *Spo12821*, which was located in the interval 1,212,661 bp~1,219,932 bp, encoding the CC-NBS-LRR protein.

To further verify whether *Spo12821* was the key gene of *RPF2*, the full sequence of the *Spo12821* gene was amplified between Sp39 (rr) and Sp62 (RR). Primers were designed according to the *Spo12821* genome sequence published in SpinachBase (http://www.spinachbase.org/, accessed 25 June 2020). The primers 12821-003 and 12821-841 successfully amplified the full sequence (Table 3). The comparison of the *Spo12821* gene between parents showed that more than 530 single nucleotide polymorphisms (SNPs) and 4 large insert/deletions (InDels), which were larger than 50 bp, and the largest InDel at about 990 bp was identified (Appendix A).

The CDS sequence of *Spo12821* was translated into an amino acid sequence. The Spo12821 encoded 1183aa and 1029aa between parents, respectively. However, the sequence of *Spo12821* was 1275aa from Sp75, suggesting extensive genetic diversity of *Spo12821*. Amino acid sequence alignment showed that Spo12821 from Sp62 share 63.86% and 72.38% sequence identity with *Spo12821* from Sp39 and Sp75. A large number of corresponding amino acid changes, insertions, and deletions were observed.

### 2.4. Structural Difference Analysis of the Candidate Gene Protein

To clarify whether the sequence variation of Spo12821 resulted in the structural change, we conducted an analysis of the protein structure of Spo12821. The results showed that the protein shared a common domain (Rx_N, NB-ARC, and LRR) in Sp62 and Sp39, but the length of the domain varied. There was only one amino acid difference in the Rx_N structure and NB-ARC structure length between the two accessions. However, in the structure of LRR, there were significant differences between Sp62 and Sp39, not only in the structural positions but also in the length. The LRR domain in Sp39 incorporates three repeats with 622 amino acids, while Sp62 has one repeat with 372 amino acids (Figure 2).

### 2.5. Development and Validation of Molecular Markers for Candidate Genes

Based on the difference in sequence of *Spo12821* from Sp39 (rr) and Sp62 (RR), a codominant marker, RPF2-IN12821, with a 68 bp difference was developed at 1,219,620 bp–1,219,904 bp on chromosome 3 (Table 4). Then, we amplified the fragment from the parents and 226 BC_1_ individuals using the marker, producing 217 bp and 285 bp in length from Sp62 and Sp39, respectively. Furthermore, all the resistant individuals (110 individuals) of BC_1_ harbored the two fragments, whereas all the susceptible individuals had one fragment (285 bp) (Figure 3). The marker could discriminate between resistant/susceptible with 100% accuracy.

## 3. Discussion

Downy mildew is the most economically impactful disease of spinach. Spinach is a green leafy vegetable, and thus leaves infected with downy mildew are unmarketable, seriously affecting production [10]. As new *P. effusa* races emerge, especially in recent years, some resistant cultivars are typically compromised by new fungal races within a short period time. The aggregation of multiple disease-resistance genes to form durable resistance has become urgent. Research on spinach downy mildew resistance genes currently focuses on the locus *RPF1*, while there is a paucity of research on *RPF2*; thus, we urgently need to develop molecular markers tightly linked to the spinach downy mildew resistance locus *RPF2*.

Previous studies have shown the three spinach downy mildew resistance loci, *RPF1*, *RPF2*, and *RPF3*, were clustered at the top of the chromosome [4]. Thirteen, two, and seven molecular markers linked to *RPF1*, *RPF2,* and *RPF3*, respectively, were developed. The two molecular markers linked to *RPF2* were developed as RPF2-1 and RPF2-2. To determine the physical location of *RPF2*, the sequence of RPF2-1 and RPF2-2 primer were aligned to the reference genome (version Sp75); RPF2-2 was aligned to 2.08 Mb on chromosome 3, while RPF2-1 forward primer and reverse primer were aligned to 1.37 Mb on chromosome 3. The candidate sequences were located in the region of 1.11 Mb to 1.72 Mb on chromosome 3, smaller than the range described in a previous study, and there was a large overlap between the candidate region and the target region (Figure 4). The *RPF1* locus was located at 0.34–1.23 Mb on chromosome 3 [17] and was mapped to the positions 0.39, 0.69, 0.94–0.98, and 1.2 Mb of chromosome 3 based on association analysis [18]. The *RPF3* locus was mapped to three physical regions of chromosome 3: 0.66–0.69 Mb, 1.05 Mb, and 1.23 Mb [19]. Combined with previous studies, we found that the region of RPF2 in this study was consistent with previous studies and that *RPF1*, *RPF2*, and *RPF3* were tightly linked. The reason for the phenomenon may be due to the low recombination in the region or the polymorphisms in one gene. 

The divergence of amino acids in the domain can affect disease resistance in plants. The *Piks* allele differs from two amino acids within the integrated heavy metal-related (HMA) domain, thus breaking the recognition of the AvrPik effector of the rice blast fungus [22]. A nonsynonymous mutation in the P-loop motif of GhDSC1 leads to amino acid sequence divergence, leading to resistance to the Verticillium wilt in cotton [23]. The CDS region of *R3* has three non-synonymous SNPs that caused three amino acid changes (S394R, E722K, and L782I). One of these amino acids, S394R, located at the conserved WHD of NBS-LRR proteins, is responsible for the resistance of the Sm gene to gray leaf spot in tomatoes [24]. Resistance is mediated by two general pathways, basal defense and R-gene-mediated defense [25]. Plant disease resistance genes (R genes) encode R proteins that have been cloned in many plant species. Plant resistance genes can be grouped into different classes according to their amino acid motif organization and membrane-spanning domains [25], among which R genes are the largest type with an extracellular nucleotide-binding site (NBS) and leucine-rich repeats (LRRs). These R genes can be divided into TIR-NBS-LRR (TNL), CC-NBS-LRR (CNL), and RPW8-NBS-LRR (RNL) subclasses according to the difference in the N-terminal structure [26,27]. The genes with nucleotide-binding domains and leucine-rich repeats (NLRs) are also considered to be the fastest evolving gene family of the R genes [23]. The NLRs gene family can induce effector-triggered immunity (ETI) that often involves a form of programmed cell death called the hypersensitive reaction (HR) [28]. The NBS domain is a part of NB-ARC domain and contains three strictly conserved motifs: P-loop, kinase-2, and kinase-3a [29]. The NBS region plays an important role as a defense signal transduction switch, which can produce a conformational shift, leading to apoptosis of infected cells [30,31]. In our study, it was found that the LRR structure of the *Spo12821* gene was significantly different in both structural location and length between Sp39 and Sp62. The LRR structure recognizes pathogen effectors to trigger immune responses [32]. It has also been demonstrated that the LRR structure is involved in protein–protein recognition in plants [33]. Therefore, small changes in the amino acid sequence in the LRR may lead to a loss of LRR function, resulting in changes in plant resistance to pathogens [34]. Based on the RNA-seq analyses of transcriptomic changes in the resistant and susceptible spinach cultivars Solomon (resistant to Pfs 1–9, 11–16) and Viroflay, we found that the expression of the *Spo12821* gene was very different in the 168 h post-inoculation period (hpi) [35] (Figure 5). These above results further verify that the *Spo12821* gene is a candidate gene for resistance to downy mildew. However, due to the limit of current spinach transgenic technology, future study will enable gene edit or transgenic activity to further validate gene function and decipher genetic mechanisms underlying resistance to downy mildew.

Marker-assisted selection (MAS) has been widely used because of its fast and efficient characteristics, and the key to MAS lies in identifying the association between genetic markers and linked quantitative traits loci (QTLs) [36]. MAS is an effective method for dealing with the increasingly severe situation of downy mildew in spinach. Additionally, in MAS, efficient molecular markers can accelerate the breeding process [37]. In addition, molecular markers can be detected throughout the entire life cycle of the plant and are not affected by the external environment. Several molecular markers have been developed in many crops for assisted breeding. Two SCAR markers, UBC359620 and OPM16750, closely linked to mildew downy genes were obtained *B. oleracea* [38]. Similarly, two molecular markers, OPK17-980 and AT.CTA-133/134, were developed that were closely linked to the downy mildew resistance gene *Pp523* in broccoli [39]. The application of molecular markers commonly used in breeding has gradually changed to InDels and SNPs, as these are characterized by high polymorphism, speed, and high efficiency. In this study, the InDel marker RPF2-IN12821 can be used for the high-throughput detection of breeding materials and can discriminate homozygous-resistant and susceptible lines from heterozygotes in a low-cost method. Molecular markers for resistance genes not only have important significance for spinach downy mildew resistance breeding but can also be potentially a powerful tool for optimizing the development of the spinach industry.

## 4. Materials and Methods

### 4.1. Plant Materials

A resistant inbred line, Sp62 (resistant to Pfs 1–10 and 15), and a susceptible line, Sp39, were selected for this study. The BC_1_ population was constructed with Sp62 as the female parent and Sp39 as the recurrent male parent that was used for subsequent susceptibility identification and *RPF2* localization. The Sp62, Sp39, and BC_1_ populations were grown in a greenhouse in Beijing (40° N, 116° E). All accessions were obtained from the Spinach Research Group, Institute of Vegetables and Flowers (IVF), Chinese Academy of Agricultural Sciences (CAAS).

### 4.2. Inoculation and Genetic Analysis

The BC_1_ population were planted in the greenhouse. Seedlings with two true leaves (14–21-days-old plants) were spray-inoculated with a previously reported sporangial suspension (2.5 × 10^5^ sporangial/mL) of *Pfs* 8 [40]. Spore suspension was evenly sprayed on spinach leaves with a watering can, and the film was covered to moisturize overnight. The film was uncovered in the morning of the next day, and then normal field management was carried out. Six days later, the film was coated again at dusk, and normal management was carried out. Subsequently, seedlings were scored as resistant or susceptible, as previously described [13]. The segregation ratios of the BC_1_ populations were analyzed using a chi-square test (χ^2^) with SAS software.

### 4.3. DNA Extraction

The fresh leaves from BC_1_ were collected for DNA extraction. Total genomic DNA was extracted using the cetyltrimethyl ammonium bromide (CTAB) method [41]. The DNA quality and concentration were assessed by electrophoresis on 1.0% agarose gels and an ND-1000 spectrophotometer (Thermo Fisher Scientific, Wilmington, NC, USA). The DNA solution was diluted to 20–100 ng/μL as the working solution and stored at –20 °C for subsequent tests.

### 4.4. The Development of InDel and KASP Markers

To obtain enough molecular markers to narrow down the candidate region, the whole genome resequencing data of Sp62 and Sp39 was conducted (Illumina, San Diego, CA, USA). The fastp (v0.12.0) software was used to filter the raw data [42]. Then, the clean data were aligned with the spinach reference genome [16] by BWA (Burrows–Wheeler alignment) software (v0.7.17-r1188), BWA-MEM algorithm, with default parameters [43]. The alignment files were used to generate variant call format (VCF) files using Samtools (v0.1.19-44428 cd) [44]. InDel and SNP polymorphisms on chromosome 3 were screened from VCF files to develop InDel and KASP markers.

### 4.5. InDel and KASP Assays

The InDel markers were generated using the Primer3 program (http://bioinfo.ut.ee/primer3-0.4.0, accessed 18 April 2020). PCR was performed under the following conditions. Briefly, after an initial denaturation at 94 °C for 5 min, the amplifications were carried out with 30 cycles at a melting temperature of 94 °C for 30 s, an annealing temperature of x (the annealing temperature was determined by the different primer sequences) for 30 s, and an extension temperature of 72 °C for 30 s, followed by a final extension step at 72 °C for 7 min. All these KASP primers were designed by the LGC company (Shanghai, China). KASP was performed under the following conditions: 15 min at 94 °C; followed by ten cycles of 20 s at 94 °C, and 60 s at 61 °C (0.6 °C drop per cycle), achieving a final annealing temperature of 55 °C; followed by a further 26 cycles of 20 s at 94 °C and 60 s at 55 °C. All plates were read below 40 °C in a 7900 HT Fast Real-Time PCR System (Applied Biosystems, Foster City, CA, USA), and the data were analyzed using SDS2.3 software (supplied by Applied Biosystems).

### 4.6. Fine Mapping of the RPF2 Locus

Genotyping information of InDel and KASP markers obtained from parental line was used to construct genetic linkage maps and located the *RPF2* gene according to the location of markers in the genome. We screened for these markers that fitted the 1:1 ratio (*p* < 0.01) in the BC_1_ population, which generated the genetic maps using software MapChart v2.3 [45].

### 4.7. The Sequence and Structural Analysis of Candidate Genes

To further verify the candidate gene, *Spo12821*, we selected the parents, Sp62 and Sp39, and designed primers according to the genome sequence published on SpinachBase by using the program primer 3 plus (https://www.primer3plus.com/, accessed 25 June 2020). After 1.0% agarose gel detection, an agarose gel DNA recovery kit (TIANGEN, Beijing, China) was used to cut the gels and recover target fragments. Subsequently, the target fragments were connected and transformed with the vector using a pEASY-T5 Zero cloning kit (TransGen Biotech, Beijing, China), and the white single colonies were selected for PCR amplification and verification. The bacterial liquid sequencing was performed by the Beijing Genomics Institute (BGI, Shenzhen, China). Then, the candidate gene sequencing results were obtained after using Multalin web online tools to carry out the multiple sequence alignment (http://multalin.toulouse.inra.fr/multalin/multalin.html, accessed 26 Steptember 2020). The Expasy online web tools were used to translate the base sequence into the amino acid sequence (https://web.expasy.org/translate/, accessed 26 Steptember 2020), and Multalin was used for sequence alignment. Finally, the EBI website Interpro was used for protein structure prediction (http://www.ebi.ac.uk/interpro/, accessed 26 Steptember 2020).

## 5. Conclusions

In this study, we developed eight molecular markers on chromosome 3, based on the resequencing of parent lines. The *RPF2* gene was narrowed down to 0.61 Mb region. Within the region, only the R gene, *Spo12821*, was identified as the best candidate gene. The comparison of sequence and structure of *Spo12821* between Sp39 (rr) and Sp62 (RR) exhibited a large sequence and structure variation, especially in the LRR domain length. Based on the sequence difference, the RPF-IN12821 marker was developed. These findings could provide a foundation for the analysis of the resistance mechanisms toward spinach downy mildew, as well as guiding the breeding of spinach resistant to downy mildew.

## Figures and Tables

**Figure 1 ijms-23-14872-f001:**
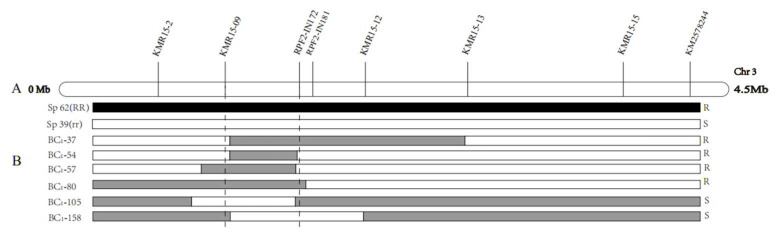
Mapping analysis of *RPF2* (**A**) Fine mapping analysis delimited *RPF2* to a 0.61 Mb interval flanked by the SNP markers KMR15-09 and RPF2-IN172. (**B**) Six recombinants are shown. The black segments represent the resistant Sp62 genotype, and the white segments indicate the susceptible Sp39 genotype.

**Figure 2 ijms-23-14872-f002:**
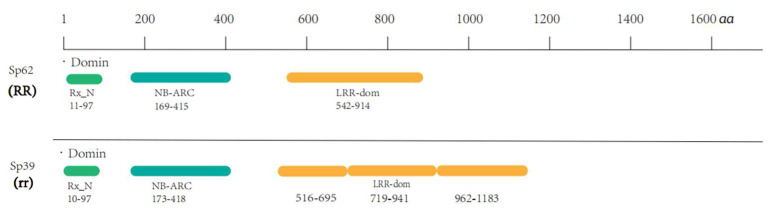
Result of Spo12821 protein structure prediction.

**Figure 3 ijms-23-14872-f003:**
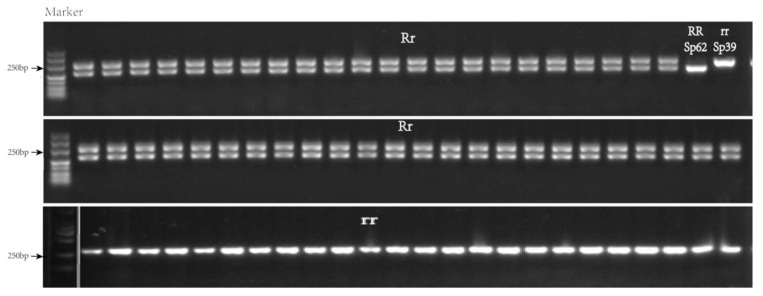
Amplification result of RPF2-IN12821 in BC_1_.

**Figure 4 ijms-23-14872-f004:**
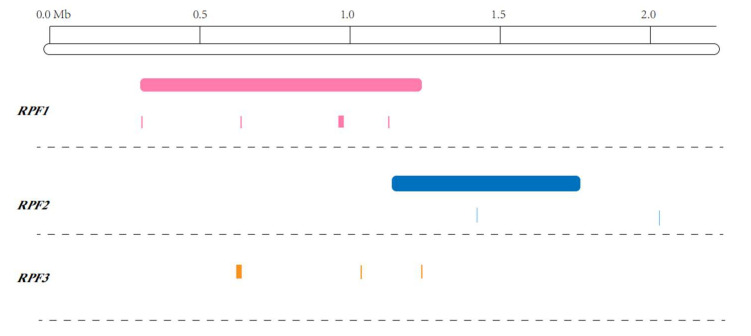
Location interval and marker position of *RPF1/2/3*. The pink segment and pink line represent the location of *RPF1*. The blue segment represents the location of our study about *RPF2*, and the blue line represents the marker position of RPF2-1 (left) and RPF2-2 (right). The orange lines represent the location of *RPF3*.

**Figure 5 ijms-23-14872-f005:**
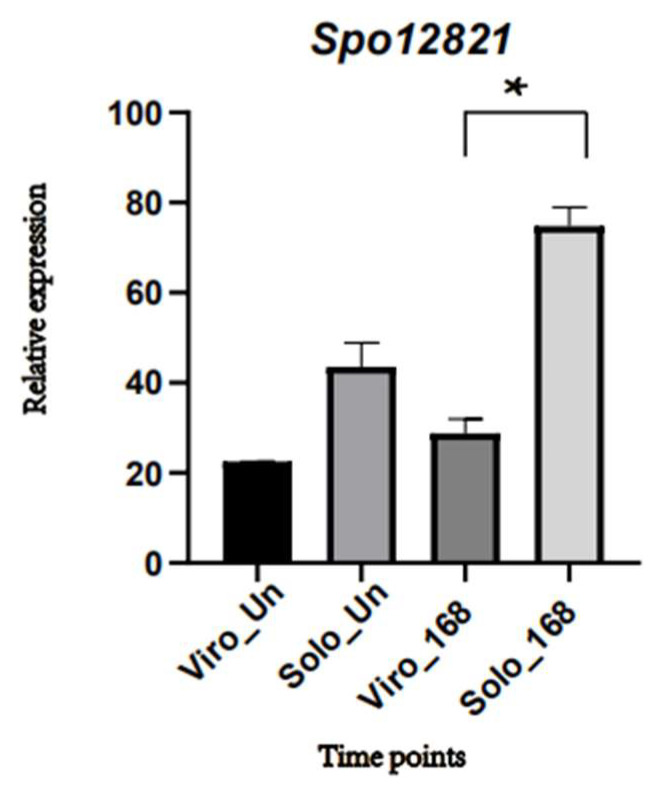
Relative expression of the gene *Spo12821* in resistant cultivar Solomon and susceptible cultivar Viroflay inoculated with Peronospora effuse at 168 h post-inoculation period (hpi) and un-inoculated control. * Represented significant difference (*p* < 0.05).

**Table 1 ijms-23-14872-t001:** Genetic analysis of resistance to spinach downy mildew in the BC_1_ population.

Population	Total Plants Number	Resistant Plant	Susceptible Plant	$\chi_{1:1}^{2}$	$\chi_{0.05}^{2}$
BC_1_	226	110	116	0.175	3.841

**Table 2 ijms-23-14872-t002:** The position of *RPF2* markers on the genome.

Chr ID	Markers	Start (bp)	End (bp)
Chr3	KM2578244	4,333,499	4,333,699
	KMR15-15	3,957,426	3,957,481
	KMR15-13	2,859,499	2,859,570
	KMR15-12	2,225,045	2,225,105
	RPF2-IN181	1,819,801	1,820,000
	RPF2-IN172	1,728,147	1,728,346
	KMR15-09	1,110,252	1,110,330
	KMR15-2	607,940	607,636

**Table 3 ijms-23-14872-t003:** Primer information of *Spo12821* amplification.

Chr ID	Markers
21-003F	GCACGTTCAGAGAAGACAG
21-003R	GGCCTTTTAGGGCTTTCAG
21-841F	GTCAAGGGGGAAGCAAGGTT
21-841R	CCGGCAGATACAGATTAAAATGG

**Table 4 ijms-23-14872-t004:** Primer information of RPF2-IN12821.

ID	Sequence (5′–3′)
RPF2-IN12821F	CTACTGATCGCCAATCTGTG
RPF2-IN12821R	CAGTCAGAAGATTTACGGCAC

## Data Availability

The raw resequencing data used in the study have been deposited in Genome Sequence Archive [46] in the BIG Data Center, Beijing Institute of Genomics (BIG), Chinese Academy of Sciences, under accession number PRJCA013482 and are publicly accessible at http://bigd.big.ac.cn, accessed on 25 November 2022.

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
