# Peer review of "Fine Mapping and Identification of a Candidate Gene of Downy Mildew Resistance, RPF2, in Spinach (Spinacia oleracea L.)"

_ijms, 2022, doi:10.3390/ijms232314872_

Round 1

Reviewer 1 Report

Dear Authors,

Below you can find some comments, I hope they could improve the soundness of you manuscript:

1. Line 28: Antioxidants are a very speculative category of compounds and have not been proven useful in large studies. spinach has a lot of useful things to mention, but not so provocative

2. Line 29: confirm with references to large studies that spinach contains substances that affect fat and carbohydrate metabolism in physiologically significant doses.

In general, there is a recommendation to remove the value of spinach as a food crop from the introduction, your article is not about that. In terms of design and data presentation, your article is flawless. Otherwise, I would really like to see real evidence of your statements according to the canons of evidence-beased medicine.

Reviewer 2 Report

Thank you for an interesting and complex study.

The authors utilized an advanced protocol to identify candidate genes of downy mildew resistance. All the sections are balanced and well-structured.

Still, the authors are recommended:

- to highlight the novelty of the study 

- to mention the limitations of the study

Reviewer 3 Report

The manuscript: “Fine mapping and identification of a candidate gene of downy mildew resistance, RPF2, in spinach (Spinacia oleracea L.)” is dealing with genetic basis of resistance against the downy mildew pathogen. According to obtained results it was verified that the Spo12821 gene is a candidate gene for resistance to downy mildew. Manuscript falls in a scope of the journal and can be accepted for the publication after minor revision.

In general, I suggest authors to  merge Result and Discussion sections in one section. There are a lot of results presented in Discussion section. All other spelling or grammar errors are highlighted in text (line 85, 124, 127, 162 and 231).
